# Improving the Performance of a Salt Production Plant by Using Nanofiltration as a Pretreatment

**DOI:** 10.3390/membranes12121191

**Published:** 2022-11-25

**Authors:** Marian Turek, Krzysztof Mitko, Paweł Skóra, Piotr Dydo, Agata Jakóbik-Kolon, Aleksa Warzecha, Klaudia Tyrała

**Affiliations:** Faculty of Chemistry, Silesian University of Technology, ul. B. Krzywoustego 6, 44-100 Gliwice, Poland

**Keywords:** coal mine wastewater, evaporated salt production, nanofiltration

## Abstract

The Dębieńsko plant in Czerwionka-Leszczyny, Poland, producing evaporated salt from the saline mine water, faces increasing operating costs due to its high energy consumption. To improve the performance of the plant, a two-pass nanofiltration with intermediate crystallization of gypsum was proposed as a pretreatment. Based on the results of pilot-scale research, it was found that the removal of most of the calcium, magnesium, and sulfate allows a substantial reduction in the concentration of these components in the concentrated brine, which is then directed to a sodium chloride crystallization evaporator. This makes it possible to increase salt yield from the current 58.8% to 76.1% and indirectly reduce energy consumption from 1350 kWh/t to 1068 kWh/t. At the same time, the volume of the highly saline post-crystallization lyes is decreased by 66%, and a new stream is obtained: a Mg-rich solution, which could be used for magnesium hydroxide recovery.

## 1. Introduction

The discharge of saline wastewater is a serious issue in many different industries. An increase in salinity in water bodies creates major environmental problems, and a radical reduction in salt discharge can only be achieved by the production of evaporated salt from these waters. To achieve this, research has been conducted on so-called zero liquid discharge (ZLD) integrated and/or hybrid systems, which use several unit operations to process the saline streams. For instance, an integrated nanofiltration (NF)-reverse osmosis (RO)-membrane crystallization system with an overall recovery of 90% was proposed [1]. In later work, the introduction of NF and a two-stage membrane crystallization was presented as a way to increase plant recovery up to 92.8% [2]. Another solution may be a hybrid system, in which the RO retentate is treated by electrodialysis (ED) [3]. The ED diluate is then recycled back to the RO feed, whereas the ED concentrate is used for the mass crystallization of salt. Other proposals include: a similar system, in which NF-RO brine is treated by ED, after which the diluate is fed to the second stage of RO [4]; a NF-ED-RO system, in which the RO retentate is mixed with the fresh seawater and fed to the NF module [5]; an integrated RO-ED-evaporation system, in which seawater RO brines are used for the production of edible salt [6]; NF-ED-RO-evaporator-crystallizer system for coal mine water [7]; and many more.

In Poland, there is one installation in operation which produces salt evaporated from saline mine waters, with an annual capacity of 100,000 tonnes [8], commonly known as Dębieńsko. It is run by Przedsiębiorstwo Gospodarki Wodnej i Rekultywacji S.A. (PGWiR), which is, in itself, a part of the Jastrzębska Spółka Węglowa (JSW) Group, a coal mining company. Recently, a detailed description of the current operation of Dębieńsko was presented by Tsalidis et al. [9]. To recap shortly: the brackish water from the "Budryk" mine and the brines from the "Budryk" mine are treated. Chemical treatment is not used. Brackish water is pre-concentrated by RO. Then the RO retentate, together with the brine, is concentrated to a NaCl concentration of approximately 290 g/dm^3^ in a vapour compression (VC) evaporator; electricity consumption is 44 kWh per ton of condensate. The brine, with a NaCl concentration of 290 g/dm^3^, is directed to the VC crystallization evaporator, where the electricity consumption is 66 kWh per ton of condensate. The largest factor limiting the salt recovery in this technology is the presence of bivalent impurities, as the crystallizer used has a hard limit on how much calcium chloride and magnesium chloride it can tolerate during the process. This causes a relatively high volume of highly saline post-crystallization lyes, which generates a secondary environmental issue. 

In this paper, an integrated system consisting of NF-RO-evaporation—crystallization is proposed. In this system, the concentrate from the evaporation method is further concentrated and, in addition to the desalinated water, the evaporated salt is obtained. The proposed system has a huge advantage from the point of view of modernization of the Dębieńsko plant, in that the company has operated the RO-evaporation-crystallization part for many years and already has the skill and equipment to continue to do so. So, for the purpose of this paper, only the NF part has been investigated experimentally on the pilot scale, while the RO-evaporation-crystallization was simulated based on empirical indicators of the industrial-scale plants. The goal of the research was to check if retro-fitting NF would show any significant improvement in terms of energy consumption, recovery, and the volume and quality of generated waste. 

The application of a NF pretreatment decreases the overall recovery of the whole system, as part of the feed water ends up in the NF retentate. Therefore, it is crucial to keep the NF permeate as high as possible; for this reason, a system of two-pass nanofiltration with intermediate gypsum precipitation was proposed, so that as much of the NF retentate can be recycled without the threat of Ca^2+^ and SO_4_^2−^ accumulation in the system leading to problems with gypsum scaling.

## 2. Materials and Methods

In the proposed system, it is necessary to conduct NF at a high stream and high yield permeate. Such conditions are achieved by using a membrane module equipped with an original spacer ensuring high mass transfer coefficient at low flow resistance—see Figure 1. The spacer is also characterized by a small variance of the residence time distribution so that it is possible to work in conditions of calcium sulfate saturation without the use of crystallization inhibitors. The spacer in question has been patented in Poland [10,11], and the international patent is pending [12]. It was shown that the NF process can be carried out in a stable way at the permeate recovery of 84.4%. The relative saturation with calcium sulfate in NF retentate with NF was 4.8, and the corresponding time of induction of nucleation was 53.5 s [13]. However, the research was conducted on a bench scale, so for the purpose of verification, the possibility of achieving high supersaturation was also tested in the pilot scale.

### 2.1. Pilot-Scale NF

The NF was tested in the pilot-scale plant consisting of 3 hydraulic stages: the first one, a commercial 2540 module (2.5" diameter, 40" long), was installed in the 1st pressure housing, while the second and third stages, each containing one 2540 module equipped with original spacers [10,11,12], were installed in the 2nd pressure housing. The plant was equipped with a calcium sulfate dihydrate crystallizer (a continuous stirred tank reactor of hydraulic residence time of 10 minutes, seeding with 150 g/dm^3^ suspension of CaSO_4_·2H_2_O), operating on the NF retentate, and a CIP chemical membrane cleaning system. The arrangement of NF and precipitation is a practical implementation of the author’s patent [14]. See the Appendix A for the process flow diagram of the plant.

NF tests were conducted on brackish water from the Budryk coal mine. The plant used the same pretreatment as is used in the RO unit in the Dębieńsko plant: sodium hypochlorite and aluminum sulfate (coagulant) were mixed with the feed water using a static mixer and passed the sedimentation tank, followed by filtration on a two-bed, sand and anthracite filter. After the two-bed filter, sodium bisulfite was dosed (to remove residual chlorine). Water from the two-bed filtration was directed to carbon filters where additional removal of suspended solids and absorption of residual chlorine takes place. The plant could operate at pressures up to 40 bar. 

During the study, the composition of the feed water was as follows (kg/m^3^): Na^+^: 10.8–12.5, Cl^−^: 19.1–22.3, SO_4_^2−^: 0.305–0.368, K^+^: 0.129–0.155, Mg^2+^: 0.600–0.715, Ca^2+^: 0.442–0.548, HCO_3_^−^: 0.091–0.098. During plant operation at very high permeate yields, the feed was acidified to avoid the crystallization of calcium carbonate in the retentate. The feed water had a silt density index (SDI) value in the range of 3.74–5.20. A SDI value of 5 is considered the limit. Supplying the plant with water with an SDI value of approxim ately 5 made it difficult to operate at high permeate yields.

Trisep TS-40 membranes were tested in all stages; a commercial 2540 module equipped with 31 mil spacers in the first stage, while the second and third stages spacers of the original design had the thicknesses of 12.6 mil and 9.1 mil, respectively. The effective area of the membranes, in each module, was, respectively (m^2^): 2.4; 3.7; 3.7.

Determinations of anions (chloride and sulfate) and cations (sodium, potassium, magnesium and calcium) were performed on an ICS-5000 Thermo Dionex chromatograph (USA). Cations were determined on an IonPac CS-16 column (Thermo Dionex, Waltham, MA, USA) and anions on an IonPac AS-19 column (Thermo Dionex, Waltham, MA, USA). Determinations of the elements Ca, Mg, S, were also carried out by inductively coupled plasma atomic emission spectrometry (ICP-AES) on a Varian 710-ES spectrometer (Varian, Belrose, Australia) equipped with a OneNeb nebulizer (Agilent, Santa Clara, CA, USA) and a double pass glass cyclonic spray chamber. Ad hoc titration methods were also used to determine chloride (Mohr method) and calcium and magnesium (complexometric method). Bicarbonate was determined using the alkalimetric method.

### 2.2. Modelling of Large-Scale NF Pretreatment

Based on the results of the pilot-scale NF with intermediate gypsum precipitation, a composition of process streams was calculated. The following assumptions were taken into account:The feed flow rate is 1 m^3^/h. In practice, scaling up the NF capacity would increase the mass of precipitated gypsum and the volumetric flow rates of the Mg-rich stream, and the NF-NF permeate, but it should not influence the ionic composition of the streams.The feed water has the average composition of “Budryk” coal mine brackish water—21.3 kg/m^3^ as Cl^−^, 0.527 kg/m^3^ as Ca^2+^, 0.681 kg/m^3^ as Mg^2+^, 0.368 kg/m^3^ as SO_4_^2−^, 12.07 kg/m^3^ as Na^+^.The entire permeate from the 1st pass of NF is fed to the 2nd pass of NF.A major portion of retentate from the 1st pass of NF is used to precipitate gypsum. The unused portion of 1st pass retentate (Mg-rich stream) is discharged and treated as waste.The rejection coefficients of ions for the 1st pass were as follows: Cl^-^ 10%, Ca^2+^ 60%, Mg^2+^ 82.8%, SO_4_^2−^ 99.1%. These were the values obtained experimentally during the pilot phase of the research at a feed flow rate of 265 L/h, permeate recovery of 82% and hydraulic pressure of 15.5 bar.The rejection coefficients of ions for the 2nd pass were as follows: Cl^-^ 18.1%, Ca^2+^ 46.2%, Mg^2+^ 61.1%, SO_4_^2−^ 92.6%. These values were obtained in bench-scale experiments on the model solutions resembling the NF1 permeate - a laboratory-scale dead-end Sterlitech® HP 4750 Stirred Cell stainless steel membrane module equipped with cooling jacket was used. The commercial flat-sheet Trisep TS40 NF membranes were cut into circular-shaped pieces, with an effective membrane area of 14.6 cm^2^, and tested under a pressure of 30 bar.The required pressure drop in each of the NF passes was calculated using Equation (1), where TDS denotes the salinity (of feed, permeate, or retentate) in kg/m^3^. See [13] for the formula derivation.The energy consumption in each of the NF passes was calculated using Equation (2), where Y denotes the permeate recovery, and P denotes the required pressure drop, as calculated by Equation (1). See [13] for the formula derivation.Gypsum can be precipitated down to the saturation level of 138% - this was confirmed experimentally during the pilot stage of the results. Gypsum seeding is used, with no additional chemicals. It was assumed that this unit operation has no energy consumption.
P = 0.448185·(TDS_feed_ − 2·TDS_permeate_ + TDS_retentate_) + 3.5,(1)
E = 0.02724 × P × Y/(0.88·0.96) + 0.05,(2)

The calculation of NF pretreatment was performed according to the following procedure: Assume some permeate recovery in NF1 and NF2.For step one, assume no recycle—the NF1 feed water is the raw coal mine water.Calculate the composition and flow rates of all streams in the two-stage NF using respective mass balances and ionic rejection coefficients.Calculate the amount of obtained gypsum assuming final saturation after precipitation is 138%.Using appropriate mass balance equations, calculate the new composition of NF1 feed, assuming the post-precipitation stream, the NF2 retentate, and the coal mine water are mixed.Compare the differences in ion concentrations and flow rate of the new NF1 feed with the NF1 feed used in previous step—if any of those differences is higher than 0.1%, go to step 3.If the calculation has converged (i.e. all errors < 0.1%), repeat steps 3–6 with increased value of NF1 and NF2 recovery.Keep increasing the values of NF1 and NF2 recovery and repeating the calculations as long as steps 3-6 can converge to a physically possible solution (e.g. positive value of precipitated gypsum, saturation of NF retentate < 400%, positive concentration of each of the ion etc.).After the maximum realistic values of NF1 and NF2 recovery have been obtained, use Equation (1) to calculate the required pressures.

### 2.3. Modelling of RO-Evaporator-Crystallizer Technology

The RO-evaporator-crystallizer part of the technology was simulated using dedicated software written by the authors in C—see the Appendix A for the source code. The empirical correlations and border conditions used by the software were based on the indicators provided by the company operating the plant and were previously discussed in [13]. The calculation algorithm is as follows:Check if the TDS of feed water is higher than 45 kg/m^3^; if yes—go to step 6.Assume RO recovery of 0.2%.Calculate the composition of RO permeate and retentate using mass balances, RO recovery, and assumed rejection coefficients [13].If Cl^−^ content is below 36 kg/m^3^, increase the RO recovery by 0.2% and go back to step 3.Calculate the required RO pressure and energy consumption [13].Using mass balance equations, calculate the composition of the process streams in the evaporator step, assuming the final Cl^-^ concentration in concentrate as 176 kg/m^3^ [13].Knowing the amount of water that needs to be evaporated, calculate the energy consumption in the evaporator.Minimize the error function of the crystallizer using the mass balance equations. The amount of evaporated water, crystallized salt and gypsum are the independent variables, whereas the maximum chloride concentration in the post-crystallization, the value of gypsum solubility product, and the bivalent cations as their respective chlorides in the post-crystallization lyes are the boundary conditions.

## 3. Results

### 3.1. Pilot-Scale

Table 1 presents the effect of permeate recovery and flow rate on the NF rejection coefficients observed during the plant run. A scaling-free operation of NF was confirmed even at very high permeate recovery (up to 92.3%)—the NF was operated continuously for 4 days at constant pressure without noticeable change in permeate flux. It was confirmed that the Trisep TS-40 membrane offers a high rejection of bivalent ions while maintaining relatively low rejection coefficients of sodium chloride—an important requirement if NF was to be used as a pretreatment in an evaporated salt production plant.

However, the scaling potential of “Budryk” coal mine brackish water was low—only 112% of gypsum saturation was obtained in the NF retentate. To test if the NF can offer stable work at a high supersaturation level, another set of experiments was performed, this time with a dosing of sodium sulfate and calcium chloride to the “Budryk” coal mine brackish water entering the NF plant. The goal was to artificially elevate gypsum saturation in the NF retentate, creating scaling conditions. This was done for two reasons: (1) to make sure the system can survive a temporary increase in CaSO_4_ concentration, which may be caused if, in the future, the coal mine decides in the future to start dewatering deeper extraction galleries (generally, the deeper the coal mine is, the more saline the water gets), (2) to make sure the system may be safely transferred to other coal mines, where the concentration of CaSO_4_ can be much higher. The results of the dosing experiments are presented in Table 2. The Na_2_SO_4_/CaCl_2_ dosing did not significantly affect the ionic rejection coefficients, but it allowed the NF retentate to reach a saturation of 300–700% with respect to gypsum. This, however, did not stop the stable NF operation—the application of thin spacers with narrow residence time distribution decreased the dead zones in the NF feed/retentate channel, meaning the supersaturated solution did not stay inside the NF module long enough to cause scaling problems. At a saturation level of 316%, the NF operated for 4 days without showing signs of scaling—at this saturation level, the gypsum nucleation induction time is 350 s.

The supersaturated retentate was used for gypsum precipitation. In a crystallization study of calcium sulfate dihydrate, it was found that, in the presence of a 150 g/dm^3^ suspension of CaSO_4_·2H_2_O (seeding method), the relative saturation was reduced from 157% to 138% with a residence time of 10 min in the crystallizer.

### 3.2. Modelling of Two-Pass NF with Intermediate Gypsum Precipitation

Figure 2 presents the results of modelling the two-pass NF with intermediate gypsum precipitation. The results suggest that such a system would achieve very high water recovery (94.8%), while simultaneously removing 53% of calcium, 80% of magnesium, and 99.7% of sulfate impurities before the final production of evaporated salt. An important feature of the proposed technology is the separation of the magnesium-rich stream (0.052 m^3^/h of 10.43 kg/m^3^ as Mg^2+^ solution). Although this was not in the scope of current research, such a high concentration of magnesium would facilitate the recovery of this valuable raw material. It is also worth noticing that the magnesium-rich waste stream is less saline (TDS of 63.03 kg/m^3^) than post-crystallization lyes, which are saturated with sodium chloride, and, thus, its discharge and disposal would be easier to handle for the company. 

It was found that the optimized conditions require hydraulic pressures of 15.5 and 18.2 bar for 1st and 2nd pass NF, respectively, which correspond to permeate recovery of 82% and 72% and energy consumption of 0.867 and 0.818 kWh/m^3^ of permeate, respectively. The calculated required pressure for both NF1 and NF2 is within typical range of NF operation. 

NF1 retentate has a gypsum saturation level of 157%, which is below the safe limits for NF operation; for this reason, the amount of precipitated gypsum was relatively low, only 0.2 kg/h. Approx. 50% of Ca^2+^ and 99.7% of SO_4_^2−^ are removed either in gypsum or in the Mg-rich stream—this is the consequence of both Ca^2+^ being in molar excess with respect to SO_4_^2−^ in the feed water (0.013 mol/dm^3^ and 0.004 mol/dm^3^, respectively) and the TS-40 membrane exhibiting higher rejection coefficient of sulphate than of calcium. The results suggest that the application of a NF membrane with higher Ca^2+^ rejection coefficient might be beneficial in this case. Such a membrane, however, might be difficult to obtain and work with: on one hand, making the NF membrane more dense would increase the calcium rejection, but it would most likely also increase the chloride rejection, which would decrease the overall salt recovery in the plant. On the other hand, application of a positively charged NF membrane might increase Ca^2+^ rejection without increasing Cl^−^ rejection, but positively charged membranes are more susceptible towards fouling.

Overall, only 9.78% of sodium chloride present in the feed water is lost with the magnesium-rich stream, which is encouraging considering all the remaining sodium chloride can be concentrated and crystallized without the bivalent impurities.

### 3.3. Modelling of RO-Evaporator-Crystallizer System

#### 3.3.1. Current Dębieńsko Operation

The composition of process streams when no NF pretreatment is used (the same technology as the current Dębieńsko operation) is presented in Figure 3. The results show why the bivalent impurities are a big issue in the Dębieńsko technology. While the feed water carries 35.1 kg/h of NaCl (calculated based on chloride concentration), only 20.64 kg/h ends up as a product—the salt recovery is only 58.8%. Because bivalent impurities are not removed from the feed water, the crystallizer also produces gypsum. The technology generates highly saline post-crystallization lyes (4.4% by volume of the feed water), which are problematic for discharge and disposal, as the plant is located deep inland, 500 km from the nearest sea. The estimated energy consumption is 1350 kWh/t of salt produced or 27.85 kWh/m^3^ of feed water. Out of this value, RO consumes 7.7%, the evaporator consumes 74%, and the crystallizer consumes 18.3%. High energy consumption is becoming an increasingly alarming issue for the plant as electric energy prices soar.

#### 3.3.2. Projected Operation after NF Pretreatment

Figure 4 presents the composition of process streams in the RO-evaporator-crystallizer part when NF pretreatment is applied. The results show that the application of two-pass NF with intermediate gypsum precipitation increases the salt recovery—26.7 kg/h of evaporated salt (76.1% of salt recovery) is produced, whereas, without the NF pretreatment, only 20.63 kg/h of salt is generated. Because the NF pretreatment changes the ionic composition of the RO-evaporator-crystallizer feed, gypsum is no longer produced in the final crystallizer. It’s also worth noting that the volume of post-crystallization lyes decreased by 66%. The projected energy consumption is 1068 kWh/t of salt or 28.5 kWh/m^3^ of feed water, out of which 5.9% is consumed by NF pretreatment, 7.1% by RO, 65.2% by the evaporator, 21.8% by the crystallizer. While this is still high, it represents a 21% decrease in energy consumption per kg of product just by installing more sophisticated pretreatment.

## 4. Discussion

Table 3 shows the comparison of performance indicators for two cases—the current operation and the operation with NF pretreatment. From the point of view of the economics of the plant, the most important indicator is the energy consumption per unit of product. The NF pretreatment shows a clear benefit, which can be explained as a result of less Ca^2+^ and Mg^2+^ present in the crystallizer, hindering its operation. The decrease in energy consumption is 21%, which would not be a drastic change, but still a significant one. 

The effect of NF pretreatment on energy consumption is twofold: 

(1) It decreases the volume of feed saline water that enters the RO-evaporator-crystallizer part by approximately 5%—smaller energy consumption by RO & evaporator. 

(2) It removes bivalent impurities which could end up in the crystallizer—meaning more salt can be recovered before the crystallizer hits a hard limit on Ca+Mg content it can tolerate in the concentrate. 

In the current operation, for every 1 m^3^ of feed water entering the plant, the energy consumption is as follows: RO 2.15 kWh & evaporator 20.6 kWh, crystallizer 5.09 kWh. A 20.62 kg of salt is recovered from every 1 m^3^ of feed water, giving the total energy consumption of 1350 kWh/t of salt. 

If NF pretreatment is applied, for every 1 m^3^ of feed water entering the plant, the energy consumption is as follows: NF 1.7 kWh, RO 2.02 kWh & evaporator 18.6 kWh (less because some part of the feed water ends up in NF retentate), crystallizer 6.21 kWh (more because the crystallizer can evaporate more water before hitting the bivalent limit). A 26.7 kg of salt is recovered from every 1 m^3^ of feed water, giving the total energy consumption of 1068 kWh/t of salt.

The results suggest that larger savings in energy consumption may occur with the optimization of the existing evaporator technology, which is responsible for the majority of energy consumption. The improvements may include using more modern, better evaporator technology than the vapour compression unit currently in operation in Dębieńsko, or using another unit operation altogether, such as a hybrid RO-ED system, as suggested in previous research [13]. On the other hand, improving the RO-evaporator-crystallizer part would be a major investment; retrofitting a NF pretreatment is cheaper and can be done without disrupting the company’s operation too much.

The application of a NF pretreatment decreases the overall water recovery from 95.6% to 93.3% (this includes RO permeate as well as evaporator and crystallizer distillates). This is, however, not an issue in the specific case of Dębieńsko: the purpose of the whole plant is not to produce water, but to produce evaporated salt. Currently, the majority of produced water is not being sold or used but discharged to the environment. The company’s main concerns are the energy consumption and the salt recovery; as long as new technology provides significant improvements in these two performance indicators, the decreased water recovery is only a secondary issue.

It is also worth noting that the application of NF pretreatment creates a new process stream—Mg-rich wastewater. While this decreases the amount of demineralized water produced by the plant, which may be perceived as economically and environmentally questionable, it also creates a significant opportunity for recovering magnesium hydroxide, thus creating a new revenue stream for the company. Magnesium is a critical raw material, and its recovery from saline streams has been studied extensively [15,16,17,18,19]. In the feed water, the ratio of Cl^−^ to Mg^2+^ is 31.3:1, while in the Mg-rich stream the ratio is 3.8:1—meaning it is a much more convenient source of this element. The highly saline post-crystallization lyes, where the Cl^−^ to Mg^2+^ ratio is 12.9:1 or 21.8:1, depending on the configuration, could, in theory, be used for magnesium recovery, but, in practice, it would be difficult because of the high salinity and density of this stream.

An important effect from the environmental point of view is the decrease in the volume of saline post-crystallization lyes. The saline wastewater needs to be diluted before it can be safely discharged into the environment. Application of NF pretreatment would increase the salt recovery from 58.8% to 76.1% and decrease the volume of lyes by 66%, meaning the salt production would not only be less wasteful, but the waste it generates would be cheaper to dispose of, and it would be less of an environmental concern.

## 5. Conclusions

Based on the results of the pilot-scale NF, the effect of NF pretreatment on the RO-evaporator-crystallizer system used for evaporated salt production was modelled. The results show that using a two-pass NF with intermediate gypsum precipitation as a pretreatment can improve the operation in several ways: -it can decrease the energy consumption by 21%, which is a positive outcome from both the economic point of view (decrease in the operating costs for the salt production plant) and from the environmental point of view (less electric energy consumption in a country where the energy grid is largely based on black coal),-it can decrease the volume of highly saline post-crystallization lyes by 66%, which makes them easier and less water-consuming for safe discharge into the environment,-it can increase salt recovery from 58.8% to 76.1%, which would help the company increase its production and decrease the amount of salt that ends up discharged to the surface waters,-it can generate a separate Mg-rich waste stream, which can be used for magnesium hydroxide recovery—this could create a new revenue stream for the coal mines and could help in transitioning from a linear to more circular economy.

The results also suggest a space for future improvements in recovery of salt from coal mine waters, namely: (1) improvements in membrane technology to produce a NF membrane with simultaneously higher Ca^2+^ rejection coefficient and lower Cl^−^ rejection coefficients, (2) improvements in thermal methods, which would decrease the energy consumption in the evaporator and crystallizer steps of the process, as those are the two most energy-consuming parts of the plant.

## Figures and Tables

**Figure 1 membranes-12-01191-f001:**
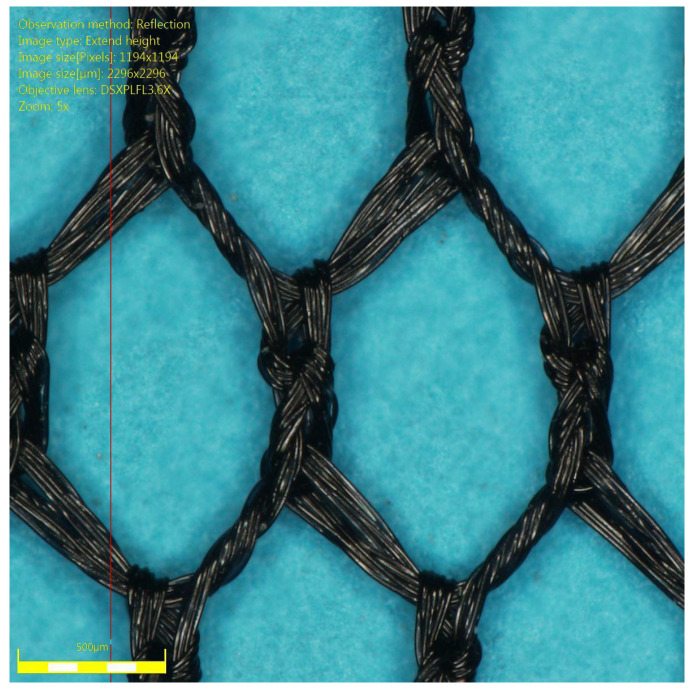
An original spacer used in this study, showing small variance of the residence time distribution and high mass transfer [10,11,12,13].

**Figure 2 membranes-12-01191-f002:**
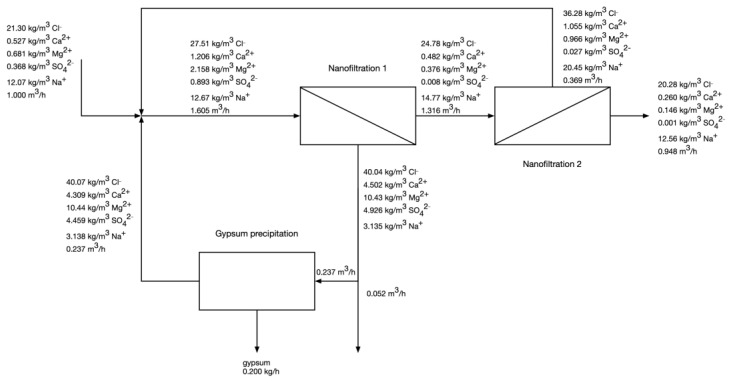
Calculated compositions of process streams in the two-pass NF system.

**Figure 3 membranes-12-01191-f003:**
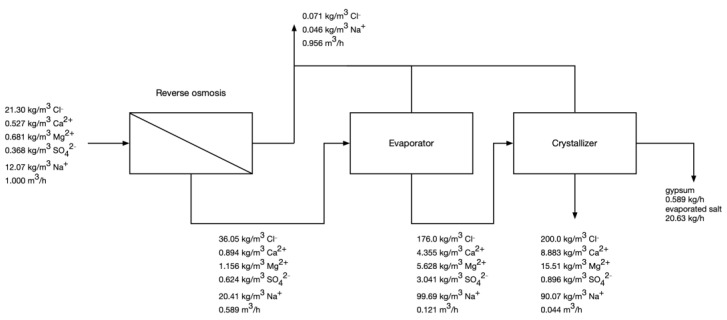
Calculated compositions of process streams in RO-evaporator-crystallizer if no NF pretreatment is used.

**Figure 4 membranes-12-01191-f004:**
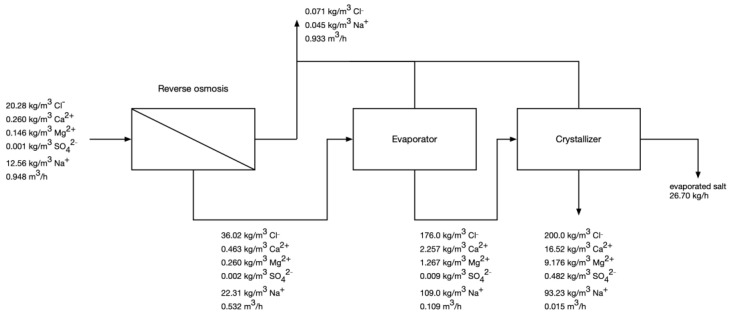
Calculated compositions of process streams in RO-evaporator-crystallizer if NF pretreatment is used.

**Table 1 membranes-12-01191-t001:** Effect of permeate recovery and the linear flow rate in the feed/retentate channel at the beginning of 1st stage/end of 3rd stage on the ionic rejection coefficients observed during NF of “Budryk” coal mine water.

Recovery [%]	Linear Flow Rate [cm/s]		Rejection [%]					
	Start	End	Na^+^	Cl^−^	SO_4_^2−^	K^+^	Mg^2+^	Ca^2+^
78.2%	11.33	5.38	9.5%	17.8%	>97.3%	9.3%	90.6%	72.0%
78.2%	11.33	5.38	8.8%	17.5%	>97.3%	9.9%	90.7%	71.8%
80.9%	19.66	5.29	6.9%	15.6%	>97.2%	10.1%	90.3%	71.1%
80.2%	20.66	5.34	7.5%	15.7%	>97.2%	8.8%	89.4%	69.3%
79.1%	20.29	5.61	8.2%	17.1%	>97.1%	10.6%	90.0%	70.4%
79.5%	20.86	5.48	9.0%	16.7%	>97.1%	9.2%	89.1%	68.4%
81.5%	6.45	2.60	4.3%	11.7%	>97.2%	4.6%	83.2%	56.2%
86.0%	20.85	3.40	8.3%	15.8%	>97.2%	8.4%	85.8%	62.5%
82.1%	18.11	3.70	7.2%	16.1%	>97.1%	7.1%	86.2%	64.4%
87.1%	7.89	2.21	3.5%	8.4%	>99.4%	6.4%	86.3%	62.4%
88.2%	12.37	1.97	4.5%	6.5%	99.4%	7.8%	85.7%	61.5%
84.7%	6.86	2.28	3.4%	10.0%	>99.1%	6.6%	82.8%	60.0%
90.8%	10.04	2.01		6.7%	94.9%		62.0%	36.8%
92.3%	12.32	2.07		5.9%	94.4%		59.0%	32.2%

**Table 2 membranes-12-01191-t002:** Concentration of Ca^2+^, SO_4_^2−^ in the feed water and the NF retentate, corresponding gypsum saturation, the NF permeate recovery and the linear flow rate in the feed/retentate channel at the beginning of 1st-stage/end of 3rd-stage during experiments with Na_2_SO_4_/CaCl_2_ dosing.

Recovery [%]	Linear Flow Rate [cm/s]		Concentration [kg/m^3^]				Gypsum Saturation [%]
	Start	End	SO_4_^2−^		Ca^2+^		
			Feed	Retentate	Feed	Retentate	
87.1	7.89	2.21	1.737	13.40	0.539	2.810	525
88.2	12.37	1.97	2.283	19.20	0.442	2.470	692
84.7	6.86	2.28	1.061	6.900	0.815	3.530	316

**Table 3 membranes-12-01191-t003:** Comparison of performance indicators between the current Dębieńsko operation (without NF pretreatment) and the operation with NF pretreatment.

Indicator	Without NF Pretreatment	With NF Pretreatment
Energy consumption [kWh/t]	1350	1068
Volume of Mg-rich stream [m^3^/h]	0	0.052
Volume of saline post-crystallization lyes [m^3^/h]	0.044	0.015
Produced salt [kg/h]	20.63	26.70
Salt recovery [%]	58.8	76.1
Produced gypsum [kg/h]	0.589	0.200
Demineralized water [m^3^/h]	0.956	0.933
Water recovery [%]	95.6	93.3

## Data Availability

Not applicable.

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
