# Peer review of "Improving the Performance of a Salt Production Plant by Using Nanofiltration as a Pretreatment"

_membranes, 2022, doi:10.3390/membranes12121191_

Round 1

Reviewer 1 Report

A few general findings are as follows:

1. Keywords should be alphabetically arranged

2.      Abbreviated words in the manuscript should be uniform e.g. use either Nanofiltration or NF (line no. 54, 63, 73, 80, 88 etc.) and Reverse osmosis or RO (line no. 45, 55, 240, 256, 273 etc.) in all places.

3.      There is no significant decrease in Ca and SO4 concentration before and after gypsum precipitation as shown in Figure 1. Please justify.

4.      Please explain the effect of using nanofiltration membrane on overall water recovery in the system as, during the nanofiltration pre-treatment, NF permeate is used for second-stage feed water that reduces the overall recovery of the system.

5.      Author stated that “using two-pass nanofiltration with intermediate gypsum precipitation as a pretreatment can improve the operation by decreasing the energy consumption by 21%”. Author needs to explain the general principle behind this reduction.

6.      Author used two-pass nanofiltration with intermediate gypsum precipitation as a pre-treatment and not stated the water recovery data in the result. Please justify.

Author Response

Thank you for the review. We believe we fixed the manuscript according to your remarks – please find responses to all the issues you raised below:

“1. Keywords should be alphabetically arranged”

Keywords have been re-arranged.

“2. Abbreviated words in the manuscript should be uniform e.g. use either Nanofiltration or NF (line no. 54, 63, 73, 80, 88 etc.) and Reverse osmosis or RO (line no. 45, 55, 240, 256, 273 etc.) in all places.”

We have revised the manuscript to use NF/RO throughout the manuscript after it was defined with the first use.

“3. There is no significant decrease in Ca and SO4 concentration before and after gypsum precipitation as shown in Figure 1. Please justify.”

The NF1 retentate already has relatively low gypsum saturation (157%) and by gypsum seeding we are only decreasing the gypsum saturation to 138%. This is not a big change, which is why the change in calcium and sulfate concentration is small. In practice, the precipitation of gypsum at low supersaturation is very slow, decreasing the saturation below 138% would require larger precipitation tank with much longer hydraulic residence time. We have chosen the 138% limit because it was what we observed in the precipitation tank we used in the pilot-scale research (CSTR having hydraulic residence time of 5 minutes, seeding with 150 g/dm3 suspension of CaSO4*2HO2).

“4. Please explain the effect of using nanofiltration membrane on overall water recovery in the system as, during the nanofiltration pre-treatment, NF permeate is used for second-stage feed water that reduces the overall recovery of the system.” and “6. Author used two-pass nanofiltration with intermediate gypsum precipitation as a pre-treatment and not stated the water recovery data in the result. Please justify.”

We think both of this issues you raised have the same justification:

We agree that the overall water recovery is decreased – it went from 95.6% to 93.3% in the overall system (including RO permeate, evaporator & crystallizer distillates), we have added data to Tab. 3 to show it explicitly. However, we believe that while the water recovery is an excellent performance indicator for nanofiltration and desalination plants in general, it is not as useful in our specific use case. The purpose of the whole plant is not to produce water, but to produce evaporated salt. Currently, the majority of produced water is not even being sold or used, but it is discharged to the local surface waters. The company running the Dębieńsko operation is selling salt and is mainly concerned with the energy consumption and the salt recovery; as long as new technology provides significant improvements in these two performance indicators, the decreased water recovery is only a secondary issue for them. We have added this discussion to the manuscript.

“5. Author stated that “using two-pass nanofiltration with intermediate gypsum precipitation as a pretreatment can improve the operation by decreasing the energy consumption by 21%”. Author needs to explain the general principle behind this reduction.”

In the Dębieńsko plant, a crystallizer is used to produce an evaporated salt; however, the crystallizer they use has a limit on how much calcium and magnesium it can tolerate in the concentrate. Currently, they don’t remove the bivalent ions from the feed water in any way whatsoever, meaning only a fraction of salt is recovered.

Application of NF works two-fold:

  • it decreases the volume of feed saline water that enters the RO-evaporator-crystallizer part by approx. 5% - smaller energy consumption by RO & evaporator,
  • it removes bivalent impurities which could end up in the crystallizer – meaning more salt can be recovered before crystallizer hits a hard limit on Ca+Mg content

We have chosen to express the energy consumption in terms of kWh of electric energy spend per tonne of salt produced, because this translates directly to the cost of the final product – salt.

In the current operation, for every 1 m3 of feed water entering the plant, the energy consumption is as follows: RO 2.15 kWh & evaporator 20.6 kWh, crystallizer 5.09 kWh. A 20.62 kg of salt is recovered from every 1 m3 of feed water, giving the total energy consumption of 1350 kWh/t of salt.

If NF pretreatment is applied, for every 1 m3 of feed water entering the plant, the energy consumption is as follows: NF 1.7 kWh, RO 2.02 kWh & evaporator 18.6 kWh (less because some part of feed water ends up in NF retentate), crystallizer 6.21 kWh (more because crystallizer can evaporate more water before hitting bivalent limit). A 26.7 kg of salt is recovered from every 1 m3 of feed water, giving the total energy consumption of 1068 kWh/t of salt.

We have added this explanation to the manuscript.

Reviewer 2 Report

Comments onmembranes

TitleTwo-pass nanofiltration in the production of evaporated salt from saline mine water

In this manuscript, based on the results of a pilot-scale nanofiltration, the effect of NF pretreatment on RO-evaporator-crystallizer system used for evaporated salt production. The results show that using two-pass nanofiltration with intermediate gypsum precipitation as a pretreatment can improve the operation in several ways: it can decrease the energy consumption by 21%, decrease the volume of highly saline post-crystallization lyes by 66%, increase salt recovery from 58.8% to 76.1%, and generate a separate Mg-rich waste stream, which can be used for magnesium hydroxide recovery. In the whole, the manuscript is relatively complete and logically rigorous, and there are some specific issues should be addressed.

1       This article focuses on two-pass nanofiltration. Please confirm whether the "UF" mentioned on line 54 is correct.

2       On line 68, it mentions that “Such conditions are achieved by using a membrane module equipped with an original spacing spacer ensuring high mass transfer coefficient at low flow resistance”. it is suggested that the authors give the diagram of original spacing spacer. This will help readers understand the purpose of the original spacing spacer.

3       On line 175, “SDI” should use the full name when it first appears.

4       On line 179, it mentions that “A scaling-free operation of NF was confirmed even at very high permeate recovery (up to 92.3%)”. Please explain which data confirms a scaling-free operation.

5       On line 207, “the beginning of 1st - state” should be revised to “the beginning of 1st - stage”.

6       On line 219, please explain why the optimal conditions for the 1st and 2nd nanofiltration are hydraulic pressures of 15.5 and 18.2 bar.

Author Response

Thank you for the review. We believe we fixed the manuscript according to your remarks – please find responses to all the issues you raised below:

“1       This article focuses on two-pass nanofiltration. Please confirm whether the "UF" mentioned on line 54 is correct.”

We have removed the UF.

“2       On line 68, it mentions that “Such conditions are achieved by using a membrane module equipped with an original spacing spacer ensuring high mass transfer coefficient at low flow resistance”. it is suggested that the authors give the diagram of original spacing spacer. This will help readers understand the purpose of the original spacing spacer.”

We have added a description of the spacer.

“3       On line 175, “SDI” should use the full name when it first appears.”

We have expanded the abbreviation.

“4       On line 179, it mentions that “A scaling-free operation of NF was confirmed even at very high permeate recovery (up to 92.3%)”. Please explain which data confirms a scaling-free operation.”

We have confirmed a scaling-free operation by running nanofiltration for 4 days under constant pressure without noticeable change in permeate flux. We have added this observation to the manuscript.

“5       On line 207, “the beginning of 1st - state” should be revised to “the beginning of 1st - stage”.

Corrected.

“6       On line 219, please explain why the optimal conditions for the 1st and 2nd nanofiltration are hydraulic pressures of 15.5 and 18.2 bar.”

We calculated the hydraulic pressures using algorithm as follows:

  1. Assume some permeate recovery in NF1 and NF2.
  2. For step one, assume no recycle – the NF1 feed water is the raw coal mine water.
  3. Calculate the composition and flow rates of all streams in the two-stage NF using respective mass balances and ionic rejection coefficients.
  4. Calculate the amount of obtained gypsum assuming final saturation after precipitation is 138%.
  5. Using mass balance equations, calculate new composition of NF1 feed, assuming post-precipitation stream, NF2 retentate and feed water are mixed.
  6. Compare the differences in ion concentrations and flow rate of a new NF1 feed with the NF1 feed used in previous step – if any of them is higher than 0.1%, go to step 3.
  7. If the calculation has converged (i.e. error < 0.1%), repeat the steps 3-6 with increased value of NF1 and NF2 recovery.
  8. Keep increasing the values of NF1 and NF2 recovery and repeating calculations as long as steps 3-6 can converge to a physically possible solution (e.g. positive value of precipitated gypsum, saturation of NF retentate < 400%, positive concentration of each of the ion etc.).
  9. After the maximum realistic values of NF1 and NF2 recovery has been obtained, use them along with the final composition of NF feed/permeate/retentate and to calculate the required pressures.

We have put this description in the manuscript.

Reviewer 3 Report

This work describes the results of a two-pass nanofiltration (NF) pilot plant with intermediate crystallization as a pretreatment method for saline coal mine water prior to evaporated salt production. The manuscript is an interesting look at how NF can be retrofitted to the existing evaporation and salt crystallization system; however, the whole process and approach lacks novelty. NF has already been proven as a suitable pretreatment method for zero liquid discharge (ZLD) integrated processes. The authors need to describe the study’s novelty and how this work can be implemented in a system outside Poland. The manuscript is generally well-written, and the results are presented in a streamlined and organized manner. I therefore recommend that this manuscript be accepted for publication in Membranes after major revisions.

Specific comments:

1.       Title: The title describes only two-pass nanofiltration; however, the study involves the simulation of the whole hybrid process aside from the pilot scale NF operation. The title feels misrepresentative of the entire work.

2.       Abstract: The motivation of this work was not presented. While the results and practical applications of the work are important, it is necessary to provide why this particular study or process was conducted, and what it hopes to contribute to an existing problem.

3.       Introduction: It was mentioned in the Materials and Methods section that the proposed system required an NF process at high stream and high yield of permeate, but it was not explained why this was required (it may be self-explanatory, but it will still be helpful for readers to know why).

4.       Materials and Methods : The pilot plant, module, and spacer used for the pilot plant should be described (2540 module by Dupont?) or shown in schematics, to provide an idea about these materials and systems

5.       Materials and Methods: Instead of presenting the composition of the brackish water feed in the Results, it should be presented in Materials and Methods.

6.       Results: Why was there a need to mimic scaling conditions at elevated gypsum concentration and longer operation periods, when the brackish feed water concentration could not induce scaling? Gypsum precipitation

7.       Results/Discussion: The methods for optimization and implications of the optimized conditions should be described, instead of just describing the permeate and retentate quality.

8.       Conclusion: It could be further expanded to include more findings and implications of the work.

Author Response

Thank you for the review. We believe we fixed the manuscript according to your remarks – please find responses to all the issues you raised below:

“1.       Title: The title describes only two-pass nanofiltration; however, the study involves the simulation of the whole hybrid process aside from the pilot scale NF operation. The title feels misrepresentative of the entire work.”

We have changed the title to “Improving the performance of a salt production plant by using nanofiltration as a pretreatment”, which we believe describes better what we were trying to achieve.

“2.       Abstract: The motivation of this work was not presented. While the results and practical applications of the work are important, it is necessary to provide why this particular study or process was conducted, and what it hopes to contribute to an existing problem.”

We have rewritten the abstract to show that we investigate the NF pretreatment to improve the performance of a Dębieńsko salt production plant, which faces increasing operating costs due to high energy consumption.

“3.       Introduction: It was mentioned in the Materials and Methods section that the proposed system required an NF process at high stream and high yield of permeate, but it was not explained why this was required (it may be self-explanatory, but it will still be helpful for readers to know why).”

We have added a short explanation to the Introduction: “The application of a NF pretreatment decreases the overall recovery of the whole system, as part of the feed water ends up in the NF retentate. Therefore, it is crucial to keep the NF permeate as high as possible; for this reason, a system of two-pass nanofiltration with intermediate gypsum precipitation was proposed, so that as much of the NF retentate can be recycled without the threat of Ca2+ and SO42- accumulation in the system leading to problems with gypsum scaling.”

“4.       Materials and Methods : The pilot plant, module, and spacer used for the pilot plant should be described (2540 module by Dupont?) or shown in schematics, to provide an idea about these materials and systems”

We have added a process flow diagram of the pilot plant to the supplementary materials.

“5.       Materials and Methods: Instead of presenting the composition of the brackish water feed in the Results, it should be presented in Materials and Methods.”

Moved to section 2.1 Pilot-scale NF.

“6.       Results: Why was there a need to mimic scaling conditions at elevated gypsum concentration and longer operation periods, when the brackish feed water concentration could not induce scaling? Gypsum precipitation”

This was done for two reasons:

  1. To make sure the system can survive temporary increase in CaSO4 concentration, which may be caused if the coal mine starts dewatering deeper extraction galleries (generally, the deeper the coal mine is, the more saline the water gets).
  2. To make sure the system may be safely transferred to other coal mines, where the concentration of CaSO4 is much higher.

“7.       Results/Discussion: The methods for optimization and implications of the optimized conditions should be described, instead of just describing the permeate and retentate quality.”

We have provided more details on optimization in the “Materials & methods”, we have also expanded discussion (new text is highlighted in yellow).

“8.       Conclusion: It could be further expanded to include more findings and implications of the work.”

We have expanded the conclusions as per your suggestion.